# Use of High-Refractive Index Hydrogels and Tissue Clearing for Large Biological Sample Imaging

**DOI:** 10.3390/gels8010032

**Published:** 2022-01-04

**Authors:** Alexander Richardson, Sandra Fok, Victoria Lee, Kerry-Anne Rye, Nick Di Girolamo, Blake J. Cochran

**Affiliations:** 1School of Medical Sciences, UNSW Sydney, Sydney, NSW 2052, Australia; Alexander.Richardson1@health.nsw.gov.au (A.R.); vicky.m.lee@gmail.com (V.L.); k.rye@unsw.edu.au (K.-A.R.); n.digirolamo@unsw.edu.au (N.D.G.); 2Katharina Gaus Light Microscopy Facility, Mark Wainwright Analytical Centre, UNSW Sydney, Sydney, NSW 2052, Australia; sandra.fok@unsw.edu.au

**Keywords:** hydrogel, tissue clearing, sample preparation, light sheet microscopy

## Abstract

Recent advances in tissue clearing and light sheet fluorescence microscopy have improved insights into and understanding of tissue morphology and disease pathology by imaging large samples without the requirement of histological sectioning. However, sample handling and conservation of sample integrity during lengthy staining and acquisition protocols remains a challenge. This study overcomes these challenges with acrylamide hydrogels synthesised to match the refractive index of solutions typically utilised in aqueous tissue clearing protocols. These hydrogels have a high-water content (82.0 ± 3.7% by weight). The gels are stable over time and FITC-IgG readily permeated into and effluxed out of them. Whilst the gels deformed and/or swelled over time in some commonly used solutions, this was overcome by using a previously described custom refractive index matched solution. To validate their use, CUBIC cleared mouse tissues and whole embryos were embedded in hydrogels, stained using fluorescent small molecule dyes, labels and antibodies and successfully imaged using light sheet fluorescence microscopy. In conclusion, the high water content, high refractive index hydrogels described in this study have broad applicability to research that delves into pathophysiological processes by stabilising and protecting large and fragile samples.

## 1. Introduction

The use of histological approaches to understand physiology and pathology has traditionally relied on optical imaging modalities. However, the opaque nature of many biological samples is a major limitation to this approach. To an extent, serial sectioning can be used to overcome this, with light scattering significantly decreased when tissues are sectioned into slices less than 100 µm thick. A major drawback to this approach is that the original 3-dimensional tissue architecture is not preserved. Whilst algorithms that automatically align serial sections can be utilised, sectioning tissue always leads to non-linear distortion. The advent of tissue clearing and whole-mount imaging systems such as light sheet fluorescence microscopy (LSFM) effectively overcomes this limitation, facilitating the visualisation of cell-to-cell interactions within complex 3-dimensional tissue architectures. Tissue clearing techniques achieve transparency via the removal of lipids and/or increasing the refractive index of the sample. Tissue clearing techniques that effectively reduce light scattering [1,2] include aqueous-based methods such as CLARITY [3,4], CUBIC [5,6,7], TDE [8], SeeDB [9], Scale [10] and MYOCLEAR [11] and organic solvent based-clearing such as BABB [12], DISCO [13,14] and PEGASOS [15]. Combined with LSFM, an unobstructed view of a whole intact tissue can be obtained [16]. The use of tissue clearing and LSFM has grown substantially in recent years and is now a widely utilised research tool in biomedical research [17]. Despite these advances, a number of challenges remain. 

Tissue clearing combined with LSFM allows for the visualisation of intact biological tissues without the requirement of tissue sectioning [18]. One important consideration when conducting LSFM is sample mounting, with agarose and phytagel being commonly used. However, samples are typically limited to less than 3 mm in order to reduce image distortion caused by refractive index mismatching between the gel scaffold and imaging media. Glues, hooks and customised holders are routinely used to mount whole, large samples such as mouse brains, but this can damage the sample. Glues can also cause light scattering where the tissue is in contact with the adhesive. To date, only a limited number of studies have investigated the utility of alternative methods of sample mounting for LSFM imaging of large biological samples.

Herein, we generated hydrogels with a refractive index that matches the CUBIC tissue clearing solutions [19] and improves sample stability and the handling of cleared tissues. Our findings show fluorescence-conjugated antibodies can freely diffuse in and out of these gels. Finally, we demonstrate that large biological samples, including mouse embryos and whole organs, can be embedded in the hydrogels, stained and successfully imaged by LSFM. This work has broad applicability to both physiological and pathological research at the whole-organ level in humans and model organisms and has the added advantage of stabilising and protecting samples.

## 2. Results and Discussion

### 2.1. Synthesis of High Refractive Index Hydrogels 

Hydrogels composed of acrylamide (Figure 1A), methacrylamide (Figure 1B) and tri(ethylene glycol) dimethylacrylate (TEDA) (Figure 1C) [19] were synthesised and evaluated for size changes (Figure 1D) and water content (Figure 1E). When immersed in distilled water, the hydrogels swelled significantly after 5 h (*p* < 0.01), 15 h (*p* < 0.005) and 24 h (*p* < 0.05) (Figure 1D). This is consistent with the uptake of water into the gel. Subsequent analysis found that the water content of the equilibrated gels was 82.0 ± 3.7% (Figure 1E), similar to previous investigations [19]. 

### 2.2. Penetration of Antibodies and Stability of High Refractive Index Hydrogels 

To determine whether hydrogels were permeable to fluorescently labelled antibodies, they were immersed in a solution of IgG-FITC for 7 days (Figure 2A). Green fluorescence reached a plateau by Day 5 and 567.8 ± 74.34% above baseline at 5 days (*p* < 0.0001) (Figure 2A). A second group of gels were immersed in IgG-FITC solution for 6 days, then washed in TBS-T to determine if IgG-FITC eluted from the gel (Figure 2B). Green fluorescence was lost in a time-dependent manner, reaching 7.74 ± 4.1% of what was observed prior to the first wash (Figure 2B) (*p* < 0.0001).

### 2.3. Hydrogels Change Size and Shape in Different Refractive Index Solutions

As tissues shrink and expand when they are cleared with CUBIC solutions, it was important to establish whether these solutions affect the hydrogels prior to tissue embedding [6]. We synthesised 1 cm^3^ hydrogels and immersed them in distilled water, CUBIC 1, CUBIC 2 or RIMS for up to 7 days (Figure 2C) and tracked changes in size over time. Hydrogels that were immersed in water swelled significantly at all time points (*p* < 0.05) (Figure 2D). Maximal swelling was reached at 8 h (112.6 ± 4.7% vs. 0 h) (Figure 2E). When immersed in CUBIC 1, a small but significant (92.4 ± 4.7%; *p* < 0.05 vs. 0 h) decrease in gel size was noted at 8 h (Figure 2E). However, by 7 days, the gel had swelled significantly (118.8 ± 7.5% vs. 0 h; *p* < 0.0001) (Figure 2E). While hydrogels that were placed in CUBIC 2 significantly decreased in size at 4 h (89.0 ± 6.8% vs. 0 h; *p* < 0.01), 8 h (89.6 ± 5.3% vs. 0 h; *p* < 0.01) and 24 h (91.9 ± 2.5% vs. 0 h; *p* < 0.05), they returned to the original size after 7 days (100.9 ± 8.2% vs. 0 h) (Figure 2F). However, these gels cracked, which could lead to impaired image acquisition. As gels that were immersed in RIMS did not change in size or fracture over 7 days (Figure 2G), this solution was selected/chosen as the imaging media for subsequent studies.

### 2.4. Stable Imaging and 3D Reconstruction of Optically Cleared, Whole-Mount Mouse Organs

“Proof-of-concept” imaging was performed during which we embedded cleared mouse tissues (Figure 3A,B) in hydrogels, immuno-stained them with Hoechst 33342 that stains nuclei, anti-αSMA, which detects blood vessels, and lectin, a general dye that binds to glycoproteins, then acquired fluorescent images using LSFM (Figure 3 and Figure 4). Using these reagents, we accurately reconstruct whole-mount E12.5 embryos and identified various anatomical features such as the paws, tail, sinuses, vessels and internal organs such as the liver (Figure 3C–E). A high degree of co-expression was observed between lectin (red) and αSMA (green) (Figure 3E). Older embryos (E14.5) were also stained for lectin and αSMA and the whole sample was rendered following image acquisition (Figure 3F and Appendix A). A high degree of detail became visible, including the developing cranial and ocular blood supply, as well as in the limbs and internal organs such as the liver (Figure 3F and Appendix A). When only the head was examined, it was apparent that lectin (red) was bound to the sinus cavities, while αSMA expression was localised to numerous blood vessels throughout the head and facial region (Figure 3G–I). 

Cleared hearts (Figure 4A–F) and kidneys (Figure 4G–I) were also imaged. An accurate reconstruction of the vasculature, including the aorta and vena cava was generated when the tissue was counter-stained with Hoechst (Figure 4A–C). Moreover, the auricles, together with their blood supply were visible when the tissue was stained with both lectin and αSMA (Figure 4D–F). Whilst an adult mouse kidney was too large to be imaged as a whole tissue using the commercial microscopy system available in this study, certain anatomical features were visualised including the ureteric artery (Figure 4G), nephron loops (Figure 4H, closed arrows) and internal vasculature (Figure 4H, open arrows) which could be rendered using a volumetric segmentation (Figure 4I). Overall, these results demonstrate the viability and applicability of utilising large, hydrogel-embedded biological samples for investigating developmental and pathophysiological scenarios.

**Figure 3 gels-08-00032-f003:**
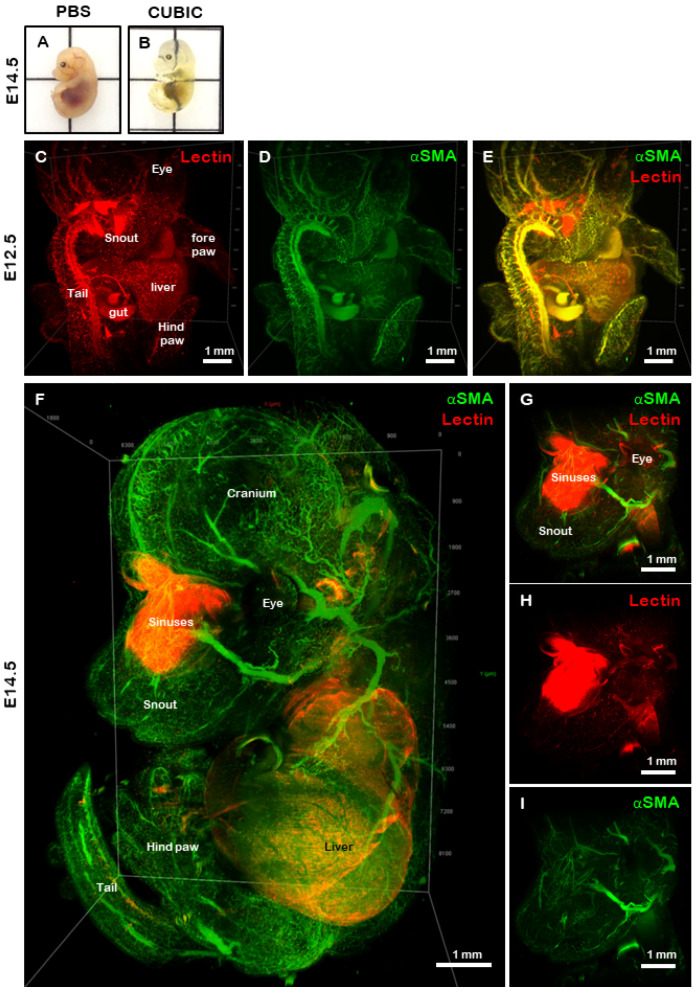
Light sheet fluorescence microscopy of CUBIC cleared mouse embryos. Representative images of an E14.5 embryo prior to (**A**) and following (**B**) clearing with the CUBIC protocol. To determine whether large biological samples could be embedded in high refractive index hydrogels prior to imaging, we stained and embedded mouse embryos at E12.5 (**C**–**E**) and E14.5 (**F**–**I**) ages. Embryos were stained with lectin (**C**,**G**) and αSMA (**D**,**H**) and the images were merged (**E**,**F**,**I**). Various anatomical features were visible on both E12.5 and E14.5 embryos. Scale bars in all panels represent 1 mm.

**Figure 4 gels-08-00032-f004:**
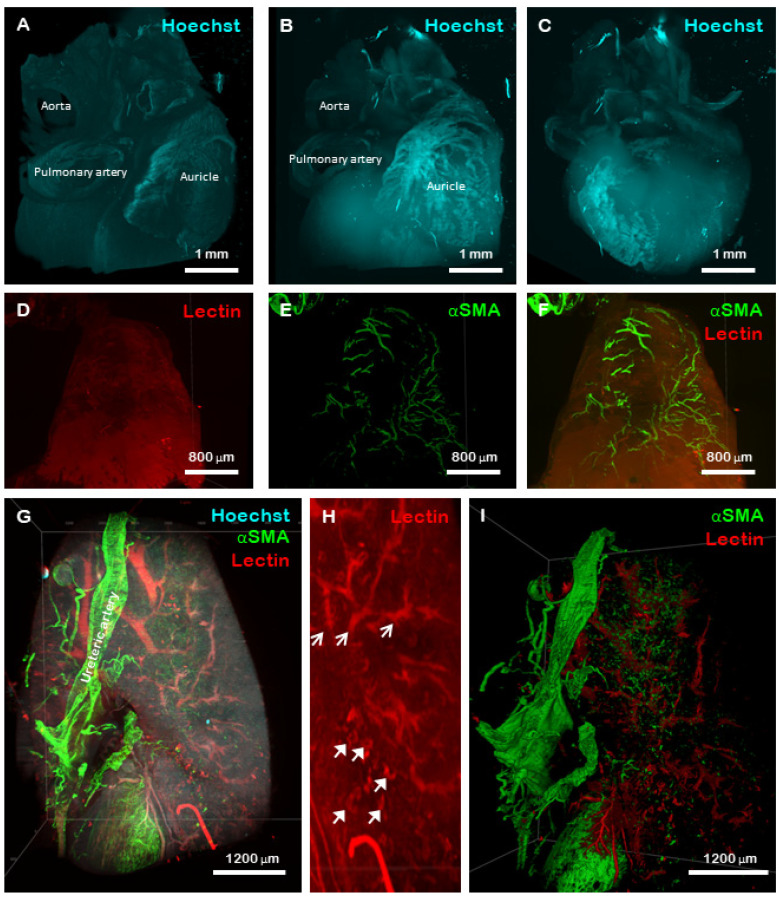
LSFM of CUBIC cleared mouse heart and kidney. A volumetric representation of a mouse heart following staining with Hoechst 33324 nuclear stain demonstrating various anatomical features is shown in (**A**). The same portion of heart tissue as a maximum intensity projection displaying the same anatomical features is shown in (**B**,**C**). Mouse auricle stained with lectin (red) and αSMA (green) showing the blood supply of the tissue is shown in (**D**–**F**). Mouse kidneys were also stained with Hoechst 33342 (blue), αSMA (green) and lectin (red) (**G**), and clearly demonstrate the ureteric artery on the outside of the capsule. When stained with lectin, individual Loops of Henle were visible within the cortex of the kidney (**H closed arrows**) together with acuate arteries and venules (**H open arrows**). A volumetric representation of the vasculature of the murine kidney stained with lectin (red) and αSMA (green) is shown in (**I**).

### 2.5. Discussion

Hydrogels are defined as a gel containing water in a three-dimensional cross-linked polymeric network formed with one or more monomers. Based on environmental changes, hydrogels can absorb or release fluids. The ability to synthesise hydrogels with tuneable physical and chemical characteristics makes them a favourable material for use in tissue engineering, ophthalmology, wound-dressing and various other biomedical applications [20]. Herein, we evaluated the applicability of a high refractive index hydrogel that provided stability to tissue architecture and minimised shrinking/expanding of the tissue for use in imaging large biological samples with LSFM [6,21]. Typically, hydrogels have either a high refractive index or high-water content but not both. High water content in hydrogels allows for efficient diffusion of drugs, peptides, proteins, oxygen and metabolites, whereas a high refractive index is desired for imaging where matching of the refractive index of amino acids and proteins (1.53–1.54) is important [22]. When the refractive index of tissue, surrounding fluid and hydrogel are matched, Snell’s law applies where the speed of light does not change when entering different media, resulting in no reflection and refraction, thereby rendering the whole unit transparent. The hydrogels utilised in this study have both a high refractive index and high water content, resulting in an ideal material for use in both imaging and diffusion studies [19]. 

Movement or retardation of proteins throughout the porous network of a hydrogel is determined by several factors including electrostatic force, hydrophobic interactions, temperature and pH [23]. In this study, FITC-labelled IgG (~13 nm diameter) was found to efficiently diffuse into and out of the hydrogel. This compatibility of the hydrogel with antibodies is important for the detection of specific cell or tissue types by immunolabelling, which together with tissue clearing can render visible the internal structures of large biological samples. Collectively, these results demonstrate that fluorescently labelled antibodies penetrated into and can be eluted from high refractive index hydrogels and that they do not non-specifically bind to gel components. This further indicates that these gels are amenable to immune labelling after tissue embedding. Furthermore, as the gels are structurally stable over time, the configuration of tissues that are embedded in them will not change, which enables repeat imaging.

A high refractive index resin has been used to match the refractive index and protect samples that have been cleared with organic solvent from oxidation, handling and preservation of GFP fluorescence in mouse brains [24]. Low-cost polydimethylsiloxane cuvettes with comparable optical properties to quartz have also been used to mount whole mouse brains for LSFM [25]. We have improved on these approaches by utilising a high refractive index hydrogel for sample stabilisation over long periods, enabling immuno-staining of large samples and repeated staining of the same sample without compromising tissue architecture. We also found that the improved structural integrity of samples embedded in the high refractive index hydrogel decreased the likelihood of structural deformation due to the external movement of the sample holder through the viscous imaging media. This resulted in improved tiling, stitching and co-registration of samples that required multiple acquisitions. This improved structural integrity of samples may be of benefit in the imaging of organoids or organs that are challenging to image such as lung or brain.

While this method was validated for intact mouse embryos and portions of both heart and kidney, the limiting factor in our ability to image larger samples was the resident microscope, which has a small imaging chamber and limited X, Y, Z movement. However, with the appropriate hardware, the method of maintaining tissue architecture using the high water content, high refractive index hydrogels described herein, will allow image acquisition of even the largest samples, including those from humans [26]. 

## 3. Conclusions

This method is applicable for researchers studying a multitude of fields and could be of use for especially large or fragile samples such as whole brains or vasculature, or to perform repeat staining of fresh or stored samples, such as clinical pathology samples, for follow up imaging [21]. Whilst this research used the CUBIC tissue clearing system, it is likely that this protocol would also be suitable for use with other tissue clearing protocols.

## 4. Materials and Methods

### 4.1. Hydrogel Synthesis

High refractive index hydrogels were synthesised as previously described [19]. In brief, acrylamide (Sigma-Aldrich, St Louis, MO, USA) and methacrylamide monomers (Sigma-Aldrich) (3:7 ratio) were prepared in MilliQ water and cross-linked with 9% (*w*/*v*) tri(ethylene glycol) dimethylacrylate (TEDA) (Sigma-Aldrich). The composition of the mixture prior to polymerisation was 1:1 (*w*/*w*) monomers:water. The solution was heated to 60 °C for 20 min to completely dissolve the monomers, then cooled to room temperature (RT), followed by the addition of 2,2′-azobis[2-(2-imidazolin-2-yl)propane] dihydrochloride (0.05% (*w*/*v*)) (Wako Chemicals, Minato City, Japan), a temperature-dependent cross-linking initiator. The solution was poured into custom-made polydimethylsiloxane (PDMS) moulds, and a thin layer of mineral oil (Singer, La Vergne, TN, USA) was applied to create an anoxic environment prior to incubation at 37 °C for 1 h. The polymerised gels were removed from the moulds and immersed in PBS for 2 days, with at least 2 changes of PBS to remove non-crosslinked monomers.

### 4.2. Water Content

Water content of hydrogels was measured as previously described [19]. In brief, samples (*n* = 8) were immersed in distilled water and equilibrated for ~2 days, then weighed on a 5-point analytical balance. The gels were then freeze-dried and weighed again. Water content was determined using the following formula:Water content (%) = ((mass_hydrogel_ − mass_dry_)/mass_hydrogel_) × 100(1)

### 4.3. Antibody Penetration

Hydrogels (*n* = 6/time point) were immersed in a solution of fluorescein-conjugated human IgG (IgG-FITC, final concentration 1 mg/mL) (Sigma-Aldrich) in TBS containing 0.1% (*v*/*v*) Triton X-100 (TBS-T). Antibody penetration was observed from wide-field fluorescence images (excitation 470/22 nm, emission 510/42 nm) that were obtained at 24 h intervals over 6 days. Antibody diffusion out of the gel was evaluated in a second set of gels (*n* = 6/time point) that had been immersed in IgG-FITC for 6 days. The gels were then washed in TBS-T, which was changed daily and images were acquired using wide-field fluorescence (excitation 470/22 nm, emission 510/42 nm). All gels were bisected prior to imaging to facilitate the visualisation of the central region. Fluorescence intensity was analysed at three regions within the centre of the gel using ImageJ software (NIH, v1.52p).

### 4.4. Hydrogel Size Change in Different Refractive Index-Matched Solutions

Prior to selecting a refractive index-matched solution for imaging, we determined how the hydrogels react to immersion in distilled water, CUBIC solutions 1 and 2 and a custom refractive index-matched solution (RIMS) [19]. CUBIC-1, comprising 25% (*w*/*w*) quadrol ([Ethylenedinitrilo]tetra-2-propanol) (Sigma-Aldrich), 25% (*w*/*w*) urea (Sigma) and 35% (*w/w*) dH2O, was stirred on a heat block set to 50 °C until all components were completely dissolved. Triton X-100 (15% (*w*/*w*, Sigma-Aldrich) was added and the solution was stirred at RT until optically clear. CUBIC-2 was prepared by stirring a mixture of 50% (*w*/*w*) sucrose, 25% (*w*/*w*) urea and 15% (*w*/*w*) MilliQ water at 80 °C on a heat block until dissolved, then cooling to room temperature. Triethanolamine (10% (*w*/*w*), Sigma-Aldrich) was added, and the solution was mixed at room temperature. RIMS was prepared with 32.4% (*w*/*v*) 60% Iodixanol (OptiPrep, Sigma-Aldrich), 29.4% (*w*/*v*) diatrizoic acid (Sigma-Aldrich) and 23.5% (*w*/*v*) N-methyl-d-glucamine (Sigma-Aldrich) in MilliQ water. Images were acquired following immersion of the samples in the hydrogel for 7 days and analysed using ImageJ software. The results are expressed as a % size change relative to T = 0 h.

### 4.5. Animals

All animal procedures were approved by the UNSW Sydney Animal Care and Ethics Committee (ACEC 17/81A). All methods were carried out in accordance with the Australian Code For The Care And Use Of Animals For Scientific Purposes (8th edition, 2013). Male and female C57BL/6 mice between 6–8 weeks of age were studied. Embryos were generated by timed mating as previously described [2]. Briefly, female mice were scented for 3 days (i.e., placed in a cage that previously housed male mice) prior to being placed with a male. The next morning the female mice were checked for the presence of a vaginal plug, which signified embryonic day (E) 0.5. All mice were euthanised by cervical dislocation following immersion in an isoflurane vaporiser. Tissues and embryos were dissected and fixed in 4% (*v*/*v*) paraformaldehyde (Sigma-Aldrich) overnight at 4 °C, then washed in PBS and stored at 4 °C.

### 4.6. Tissue Clearing

Tissues, including heart and kidney, and E12.5 and E14.5 embryos were optically cleared using the CUBIC protocol as previously described [6]. In brief, tissue was immersed in CUBIC 1 solution, which was changed every 2–3 days until the sample was visibly clear (Figure 3A,B). The samples were then washed in PBS for 3 days with at least 3 changes of solution prior to further use. 

### 4.7. Embedding Cleared Tissue in Hydrogel and Immunostaining

Cleared tissues were immersed in a hydrogel solution and allowed to equilibrate for 48 h, then cross-linked as described above. The samples were washed over 24 h with at least 3 changes of PBS, then immunofluorescently stained using Hoechst 33342 nuclear stain, an anti-alpha-smooth muscle actin (αSMA) antibody conjugated to AlexaFluor-488 (53-9760-82, Abcam; 1:100 dilution) and Tomato lectin conjugated to DyLight-649 (DL-1178, Vector Labs; 1:1000 dilution) in TBS-T containing 2% (*w*/*v*) bovine serum albumin (TBS-T/BSA) for 7 days at 4 °C. The samples were then washed in TBS-T/BSA for a further 3 days, with at least 3 changes of the buffer. Finally, the samples were immersed in RIMS to ensure the correct refractive index for image acquisition. 

### 4.8. Light Sheet Fluorescence Microscopy (LSFM)

Images of immunostained samples were acquired using a Lightsheet Z.1 microscope (Zeiss) equipped with a 5× dry objective (NA 0.1) (Katharina Gaus Light Microscopy Facility, Mark Wainwright Analytical Centre, UNSW Sydney) using the following settings: Hoechst 33342: excitation 458 nm, emission bandpass 460–500 nm; Alexafluor 488: excitation 488 nm, emission bandpass 505–545 nm; DyLight 649: excitation 638 nm, emission long pass 600 nm. A tiled series of z-stacks between 800–1000 images with 10% overlap were acquired and processed with Vision 4D (Arivis) software.

### 4.9. Statistical Analysis

Results were analysed for statistical significance using a one-way ANOVA with Dunn’s multiple comparison test using Prism 9 (GraphPad). A *p*-value *<* 0.05 was considered significant. Any other relevant statistical tests are noted in the appropriate figure legend.

## Figures and Tables

**Figure 1 gels-08-00032-f001:**
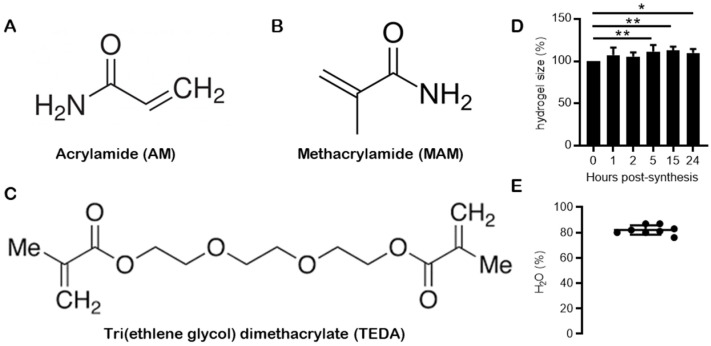
Physical description of high refractive index hydrogels. The chemical structures of acrylamide (**A**), methacrylamide (**B**) and tri(ethlene glycol) dimethacrylate (**C**) are shown. Changes in hydrogel size following synthesis and immersion in PBS for 24 h were measured (*n* = 8) (**D**), together with the water content of high refractive index hydrogels (*n* = 8, each point is an independent experiment) (**E**). Values (**D**,**E**) representing the mean ± SD and results were analysed by one-way ANOVA with Dunn’s multiple comparison’s test. * *p* < 0.05, ** *p* < 0.01.

**Figure 2 gels-08-00032-f002:**
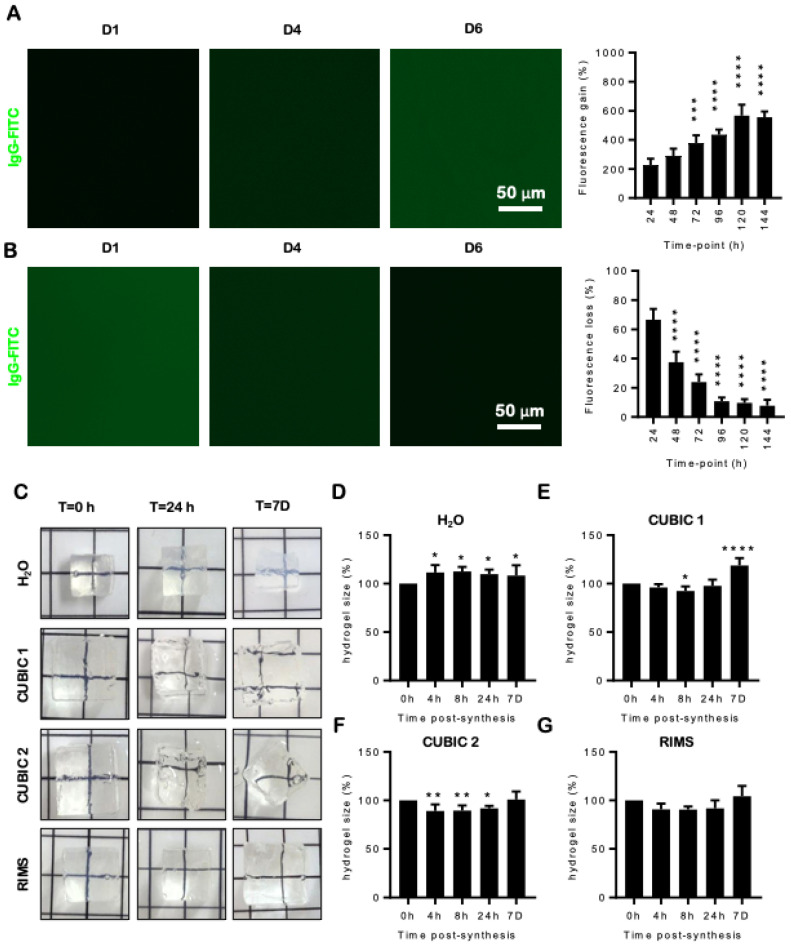
Antibody penetration, size changes in different media and high refractive index hydrogel motility over time. High refractive index hydrogels were assessed for their ability to influx (**A**) and efflux (**B**) IgG-FITC. Fluorescence intensity was quantified over 6 days. Histograms (**A**,**B**) represent mean (n=6/time point) fluorescence intensity ± SD. Hydrogels were analysed for volume changes in different refractive index matched solutions including distilled water (**C top row and D**), CUBIC-1 (**C second row and E**), CUBIC-2 (**C third row and F**) and RIMS (**C bottom row and G**). Histograms (**D**–**G**) represent mean (n = 6/time point) hydrogel volume ± SD and are displayed as the % change from t = 0. Data were analysed by one-way ANOVA and Dunn’s multiple comparisons test. * *p* < 0.05, ** *p* < 0.01, *** *p* < 0.001, **** *p* < 0.0001.

## Data Availability

Data supporting the claims made in this study are available upon reasonable request to the corresponding author.

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
