# Peer review of "Use of High-Refractive Index Hydrogels and Tissue Clearing for Large Biological Sample Imaging"

_gels, 2022, doi:10.3390/gels8010032_

Round 1

Reviewer 1 Report

The article is of interest in tissue clearing field.
It shows a potential in imaging organoids which are fragile and difficult to image in the light-sheet microscope. It would be interesting to see if hydrogels might be used for organoids as well. Did the Authors tested other samples (i.e. organoids) apart from mentioned in the article?

Tested tissues and embryos are relatively stiff. I wonder if the Authors tested (even unsuccessfully) other organs such as brain or lungs that are challenging to image after clearing using CUBIC. Please comment.

It is not clear how the samples were mounted in the microscope. Were the samples in cubes cut out from the gel? If so, were the cubes much bigger than samples and if the hydrogel cubes size influenced the imaging quality. Please provide a picture of samples embedded in hydrogel. 

Please provide more details on imaging setup used, i.e. information on objectives (magnification, NA, etc) is missing. What was the image size? Was the sample rotated and stitched in postprocessing? 

Author Response

It shows a potential in imaging organoids which are fragile and difficult to image in the light-sheet microscope. It would be interesting to see if hydrogels might be used for organoids as well. Did the Authors tested other samples (i.e. organoids) apart from mentioned in the article?

Response: We thank the reviewer for this insightful comment. We did not test the utility of the hydrogels developed in this study for use with organoids. The purpose of our study was to test the feasibility of the gels with samples embedded for staining and imaging and as such, we did not attempt sample types that would pose a technical challenge on their own in terms of samples handling. The reviewer is correct that the use of these gels could prove useful in the imaging of other sample types, including fragile samples such as lungs and brains or organoids. Considering this comment, we have added an additional sentence (line 209-210) in the discussion section of the revised manuscript.

Tested tissues and embryos are relatively stiff. I wonder if the Authors tested (even unsuccessfully) other organs such as brain or lungs that are challenging to image after clearing using CUBIC. Please comment.

Response: See previous comment. Whilst we agree that this would be interesting to try, as this study is a proof-of-concept study on the use of the high refractive index hydrogels, we wanted to try tissues that would pose less of a technical challenge in terms of sample handling.

It is not clear how the samples were mounted in the microscope. Were the samples in cubes cut out from the gel? If so, were the cubes much bigger than samples and if the hydrogel cubes size influenced the imaging quality. Please provide a picture of samples embedded in hydrogel.

Response: Samples were contained inside gels cast in custom made PDMS moulds, as described at line 233-234.We have added the wording ‘custom-made’ to ensure that it is clear that the moulds were designed specifically for the samples. As such, the samples were mounted inside a block of gel just bigger than the sample. We chose not to include the specific mounting steps used in this study as these are likely to vary significantly between different imaging systems and this study focused on the characterisation of the hydrogels, rather than as an imaging techniques paper.

Please provide more details on imaging setup used, i.e. information on objectives (magnification, NA, etc) is missing. What was the image size? Was the sample rotated and stitched in postprocessing?

Response: Samples were not moved or rotated during imaging, any rotation (such as the supplementary video) is purely artificial post-processing. Imaging was performed using a 5X dry objective (NA 0.1). Samples were mounted in imaging media inside a sample chamber, then imaged in a single series of images from front to back to generate a 3D image set. These were then compressed into the 2D images presented in the manuscript. The objective details have been included at line 308 of the revised manuscript.

Reviewer 2 Report

In this project, the authors transfer the research carried out by Zhang C. et al to the realm of light sheet fluorescence microscopy (LSFM). More specifically, they use hydrogels that have the usually incompatible but highly desirable properties of high refractive index and high-water content as a mounting frame for biological samples, to improve their structural stability during handling and imaging.

While commonly used clearing methods such as ACT-PRESTO, MYOCLEAR, CLARITY, SHIELD or SWITCH require the hydrogel to provide the sample with thermal and chemical stability to withstand the harsh protocol steps, in this project the authors use the hydrogel in combination with a sample-friendly optical clearing method such as CUBIC to improve, not the clearing method itself, but the handling of the sample during the experimental procedure and the imaging stage. In a fast-growing research field such as optical clearing in which most efforts are dedicated to developing more efficient and faster protocols, it is also valuable to find research devoted to aiding and optimizing the experimental procedure itself.

The manuscript is well organized and self-explicative, the experimental pipeline is conceptually straight-forward and well presented, and the results conform to the state of the art. However, there are some issues that should be highlighted. They will be addressed in the following lines for each of the sections of the manuscript.

The introduction presents sufficient background for a non-expert to understand the topic at hand. However, the motivation and objectives of the project are not clearly stated. The only mention to the true goal of the project is half a sentence in line 63, while the rest of the paragraph enumerates how the side effects of using a hydrogel have been overcome. A much more clarifying motivation is presented in the Discussion section in lines 204-212. I would strongly suggest moving the content of those lines to the introduction.

The paragraph three is confusing: it mentions the effect of several clearing methods on the mechanical and chemical properties of the sample, but the goal of this listing is not evident to me. As far as I have understood from the manuscript, the proposed method does not alter the properties of the sample, just provides and external framework to protect the sample. Additionally, most of the effects of the clearing methods do not even seem negative, and furthermore, the last 6 lines of the paragraph are devoted to organic-based solvents, which are incompatible with the strategy presented in the project.

In section 2.2 of results, the last paragraph should be moved to the Discussion section, as it contains an analysis of the results. Regarding the experimental design, it’s confusing why permeability and antigenicity of the hydrogel was evaluated with IgG-FITC, if the sample was labelled with Hoechst 33342, lectin and αSMA. Why not use one of them instead? IgG-FITC may not bind to the hydrogel but lectin or αSMA may do (In line 197 the authors state that diffusion of antibodies depends on “hydrophobic interactions”, which may be antibody specific).

Also, regarding the colocalization study, a time window of 100s does not seem enough to conclude minimal structural motility of the gel, specially given that the duration of the protocol and imaging is significantly longer. This is specially surprising given that the rest of experiments were carried out for much longer time periods.

In section 2.3, the authors say that they selected RIMS as the most appropriate imaging media because it had the less aggressive effect on the hydrogel. However, they do not mention if RIMS can properly diffuse into the hydrogel and more importantly, into the sample. This is fundamental because if the refractive indexes within the sample are not matched, the sample will not become sufficiently transparent. The authors should also mention the minimum immersion time required to maximize transparency.

In material and methods section, it would be of great help if the authors included a visual diagram of the different optical clearing protocol steps including the hydrogel synthesis and embedding and antibody labelling, comparing the duration of the pipeline with the traditional CUBIC, to allow the reader to rapidly assess the proposed protocol at a glance. Regarding the experimental pipeline itself, it’s not clear why the antibody labelling step must be done after the hydrogel embedding and not before it. None of the steps of the embedding seem prone to affect the antibody labelling, and given the amount of time that the antibody takes to penetrate the hydrogel (Figure 2), it may prove more optimal to do the antibody labelling before, unless the proposed pipeline allows for multiplexing, as it may be understood in line 218 of the Discussion. However, it is not clear whether the hydrogel would withstand elution.

It would be helpful if the authors specified if the hydrogel solution penetrates the sample as in the case of CLARITY or SWITCH or if it only encloses it as an Agarose gel.

Minor issues:

  1. In line 46, it may prove useful to add the paper by Costantini, Irene, et al. "In-vivo and ex-vivo optical clearing methods for biological tissues." Biomedical optics express 10.10 (2019): 5251-5267, to demonstrate the growth of the field.
  2. In line 53, the sentence “Scale creates cleared tissues” should be substituted by “Scale leads to cleared tissues” or similar.
  3. The authors should indicate the program used to carry out the statistical analysis.
  4. In line 275, the authors should change “CUBIC solutions 1 and 26” by “CUBIC solutions 1 and 2”.

Author Response

The introduction presents sufficient background for a non-expert to understand the topic at hand. However, the motivation and objectives of the project are not clearly stated. The only mention to the true goal of the project is half a sentence in line 63, while the rest of the paragraph enumerates how the side effects of using a hydrogel have been overcome. A much more clarifying motivation is presented in the Discussion section in lines 204-212. I would strongly suggest moving the content of those lines to the introduction.

Response: On the advice of the reviewer, we have moved lines 204-212 from the Discussion section and placed this in the Introduction. This paragraph is now lines 47-56 of the revised manuscript.

The paragraph three is confusing: it mentions the effect of several clearing methods on the mechanical and chemical properties of the sample, but the goal of this listing is not evident to me. As far as I have understood from the manuscript, the proposed method does not alter the properties of the sample, just provides and external framework to protect the sample. Additionally, most of the effects of the clearing methods do not even seem negative, and furthermore, the last 6 lines of the paragraph are devoted to organic-based solvents, which are incompatible with the strategy presented in the project.

Response: After careful consideration, we agree with the reviewer. As such, we have removed this paragraph from the revised manuscript as we do not feel that its inclusion is warranted.

In section 2.2 of results, the last paragraph should be moved to the Discussion section, as it contains an analysis of the results. Regarding the experimental design, it’s confusing why permeability and antigenicity of the hydrogel was evaluated with IgG-FITC, if the sample was labelled with Hoechst 33342, lectin and αSMA. Why not use one of them instead? IgG-FITC may not bind to the hydrogel but lectin or αSMA may do (In line 197 the authors state that diffusion of antibodies depends on “hydrophobic interactions”, which may be antibody specific).

Response: We have moved the last paragraph of section 2.2 of the previous version of the manuscript to the discussion section. It is now part of the Discussion section at lines 191-196 of the revised manuscript. We chose to use IgG-FITC in preliminary experiments due to its ready availability and low cost – our reasoning was that if a non-specific antibody could readily diffuse in and out of the hydrogel, this was also likely to be the case for other antibodies and small molecules. As water soluble reagents that are commonly utilised in fluorescence microscopy, we believe these reagents should be compatible with the imaging approaches described in this study. Indeed, these reagents did not readily leach into the hydrogel and result in the formation of significant background signal, which would be expected in the aqueous environment of the tissue-hydrogel sample if the reagents had a high affinity for the hydrogel.

Also, regarding the colocalization study, a time window of 100s does not seem enough to conclude minimal structural motility of the gel, specially given that the duration of the protocol and imaging is significantly longer. This is specially surprising given that the rest of experiments were carried out for much longer time periods.

Response: After careful consideration, we believe that this comment can only be addressed with considerable extra experiments including electron microscopy to characterise the pore size of the polymerised hydrogels. This is beyond the current capability of the research team involved in this study, and we are not currently sufficiently resourced to undertake such experiments. In light of this, we have decided to remove the experiment referred to by the reviewer in this comment We do not consider that this removal has a significant impact on the either the quality or scientific merit of the revised manuscript.

In section 2.3, the authors say that they selected RIMS as the most appropriate imaging media because it had the less aggressive effect on the hydrogel. However, they do not mention if RIMS can properly diffuse into the hydrogel and more importantly, into the sample. This is fundamental because if the refractive indexes within the sample are not matched, the sample will not become sufficiently transparent. The authors should also mention the minimum immersion time required to maximize transparency.

Response: As explained in our response to Reviewer 1, samples were not moved or rotated during imaging. As we were able to image through thick samples in such a manner without any significant distortion or compromise of image quality, we are encouraged that the RI of samples are matched and consistent throughout the whole sample.

In material and methods section, it would be of great help if the authors included a visual diagram of the different optical clearing protocol steps including the hydrogel synthesis and embedding and antibody labelling, comparing the duration of the pipeline with the traditional CUBIC, to allow the reader to rapidly assess the proposed protocol at a glance. Regarding the experimental pipeline itself, it’s not clear why the antibody labelling step must be done after the hydrogel embedding and not before it. None of the steps of the embedding seem prone to affect the antibody labelling, and given the amount of time that the antibody takes to penetrate the hydrogel (Figure 2), it may prove more optimal to do the antibody labelling before, unless the proposed pipeline allows for multiplexing, as it may be understood in line 218 of the Discussion. However, it is not clear whether the hydrogel would withstand elution.

Response: Upon request from the journal, we have submitted a graphical abstract subsequent to our submission of the manuscript. This graphical abstract is included below and covers the details raised here by the reviewer. We do not believe there is a requirement to perform the staining after the gel polymerisation, samples could be stained prior to being embedded in gel. We did not test elution of staining from the hydrogel as this study and whilst we agree that this could be useful, it is beyond the scope of the current manuscript.

It would be helpful if the authors specified if the hydrogel solution penetrates the sample as in the case of CLARITY or SWITCH or if it only encloses it as an Agarose gel.

Response: Please see comment above re RIMS.

Minor issues:

In line 46, it may prove useful to add the paper by Costantini, Irene, et al. "In-vivo and ex-vivo optical clearing methods for biological tissues." Biomedical optics express 10.10 (2019): 5251-5267, to demonstrate the growth of the field.
This reference has been included.

In line 53, the sentence “Scale creates cleared tissues” should be substituted by “Scale leads to cleared tissues” or similar.

This correction has been made.

The authors should indicate the program used to carry out the statistical analysis.

We have included the use of Prism 9 (GraphPad) for statistical analysis at line 487 of the revised manuscript.

In line 275, the authors should change “CUBIC solutions 1 and 26” by “CUBIC solutions 1 and 2”.

This correction has been made.